# Discovering covalent cyclic peptide inhibitors of peptidyl arginine deiminase 4 (PADI4) using mRNA-display with a genetically encoded electrophilic warhead
Isabel R. Mathiesen [1,2], Ewen D. D. Calder [1,2], Simone Kunzelmann [3] & Louise J. Walport [1,2] ✉

Covalent drugs can achieve high potency with long dosing intervals. However, concerns remain about side-effects associated with off-target reactivity. Combining macrocyclic peptides with covalent warheads provides a solution to minimise off-target reactivity: the peptide enables highly specific target binding, positioning a weakly reactive warhead proximal to a suitable residue in the target. Here we demonstrate the direct discovery of covalent cyclic peptides using encoded libraries containing a weakly electrophilic cysteine-reactive fluoroamidine warhead. We combine direct incorporation of the warhead into peptide libraries using the flexible in vitro translation system with a peptide selection approach that identifies only covalent target binders. Using this approach, we identify potent and selective covalent inhibitors of the peptidyl arginine deiminase, PADI4 or PAD4, that react exclusively at the active site cysteine. We envisage this approach will enable covalent peptide inhibitor discovery for a range of related enzymes and expansion to alternative warheads in the future.

Covalent inhibitors convey beneficial properties including increased potency, simpler pharmacokinetics, due to non-equilibrium kinetics, and potential for extended dosing intervals[1,2]. Additionally, covalent inhibitors are useful for competition with high concentrations of endogenous ligands due to their nonequilibrium binding mechanism. However, there remain concerns about off-target effects, due to the reaction of the covalent warhead with other proteins[3]. Targeted covalent inhibitors (TCIs) address this by combining weakly electrophilic warheads with high affinity scaffolds which optimally position the reactive group to react at a target residue[4,5].

Peptides make an ideal modality for the high affinity scaffold due to their tight binding affinities, high target selectivity and relative ease and low cost of synthesis. Having a comparably small size, they can be orally bioavailable, whilst still having a sufficiently large surface area to target relatively featureless protein interfaces with high specificity[1,6–9]. Macrocyclisation of peptides confers additional benefits including high proteolytic stability and increased potency[10,11].

Classically, covalent peptides are developed through the addition of a warhead into a previously identified reversible binder or substrate analogue, requiring structural information or laboriously generated structure-activity relationship information[12–15]. Identifying a suitable site for warhead addition that enables efficient reaction without disrupting potent target binding is challenging. Additionally, in many cases neither a substrate analogue nor structural information is available. Addressing both these challenges, direct identification of covalent peptides from high-throughput screening offers a route to speed up covalent drug discovery.

Genetically-encoded peptide screening platforms, such as phage display, mRNA display and the related random non-standard peptides integrated discovery (RaPID) system, provide powerful approaches to identify peptide hits from enormous libraries of cyclic peptides (up to $10^{13}$ sequences)[16–19]. These platforms have been used successfully to identify potent reversible chemical tools and drug candidates to a wide range of targets[7,20–23]. Recently these screening approaches have been modified to promote bias towards the discovery of irreversible covalent inhibitors; reactive moieties have been introduced into the peptide libraries alongside denaturing guanidine washes during the peptide selection step[24,25]. For example, a modified phage display protocol has been used to identify de

---

[1]Protein-Protein Interaction Laboratory, The Francis Crick Institute, London, UK. [2]Department of Chemistry, Molecular Sciences Research Hub, Imperial College London, London, UK. [3]Structural Biology Scientific Technology Platform, The Francis Crick Institute, London, UK. ✉e-mail: l.walport@imperial.ac.uk

novo covalent peptide inhibitors through post-translational modification of peptide libraries with warheads into a fixed position — either within the cyclisation linker or at reduced disulphide bonds[26–29].

Other methods to introduce unnatural chemistry into peptides have also been developed, including through use of modified aminoacyl tRNA synthetases, chemical aminoacylation of tRNA or use of aminoacylating ribozymes, known as flexizymes[17,30–34]. The RaPID system offers a route to identify chemically diverse peptide binders through encoding non-canonical amino acids in displayed peptides using the flexizyme-mediated flexible in vitro translation (FIT) system[23,35]. This enables both facile peptide cyclisation and the potential for direct warhead incorporation[36,37]. Unlike in the phage display approaches this allows incorporation of the covalent warhead at variable positions in the peptide macrocycle. Recently, this strategy has been used to incorporate phenylselenocysteine into RaPID libraries, which was then post-translationally modified to yield a dehydroalanine warhead[24]. Photoreactive covalent peptides have also been identified through the incorporation of a benzophenone moiety[25]. However, the ability to directly encode an unmasked electrophilic warhead within displayed libraries, rather than relying on post-translational modification, has not yet been exploited.

Peptidyl arginine deiminase 4 (PADI4 or PAD4) is one of five enzymes in the PADI family. PADIs 1–4 catalyse the post-translational modification of peptidyl arginine residues to citrulline in a wide range of protein substrates[38]. PADI4 is involved in cell signalling processes including apoptosis, differentiation, and regulation of transcription[39–42]. Dysregulation of PADI4 is implicated in various diseases including rheumatoid arthritis, lupus and several cancers[43–45]. PADI4 has a key active site Cys residue required for catalysis and there are known covalent small molecule binders[46,47]. Fluoroamidine (1) is a small molecule inhibitor of PADI4; as an arginine mimetic (Fig. 1A) it binds in the active site of PADI enzymes and covalently reacts with the active site cysteine (C645)[47]. However, it has limited selectivity for PADI4 over PADI1[48]. From a screen of synthetic peptides containing 1, the tripeptide TDFA was identified with an IC$_{50}$ of 2.3 μM and >15-fold selectivity for PADI4 over PADI1[49]. Based on this precedent we envisaged that developing a high-throughput methodology to identify larger covalent peptides including the fluoroamidine warhead might provide a route to even more potent and selective inhibitors.

Here we report the direct incorporation of the cysteine reactive electrophile, fluoroamidine, into cyclic peptide RaPID libraries produced by in vitro translation. We applied our covalent RaPID library in a screen against PADI4 to select exclusively for peptides that are covalently bound to the target protein. Our approach yielded peptides that covalently bind to PADI4 at Cys645 and inhibit PADI4 citrullination activity, three of which have

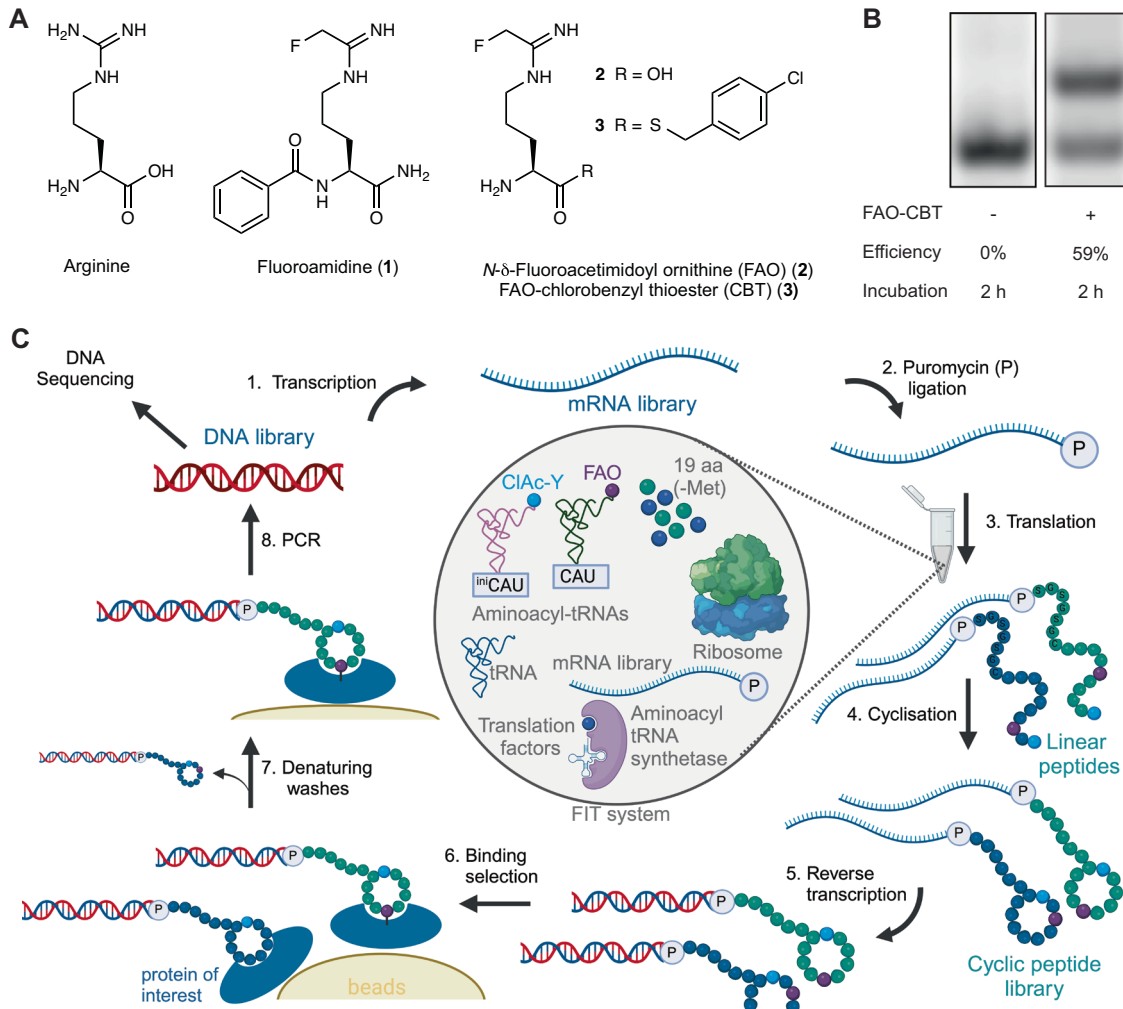

**Fig. 1 | Covalent RaPID setup. A** The structure of arginine and related arginine-mimetic PADI4 inhibitor **1** and synthesised unnatural amino acids **2** and **3**. **B** Microhelix assay using a truncated tRNA mimic to monitor loading of FAO-CBT (**3**) using eFx. The upper band indicates the presence of aminoacylated microhelix RNA and the lower band is non-aminoacylated microhelix RNA. After 2 h incubation at 4 °C, 59% aminoacylation is seen. The full gel is provided in Fig. S23. **C** The covalent RaPID cycle setup. Transcription, puromycin ligation, translation, reverse transcription, and affinity panning against immobilised PADI4 are performed as in a typical RaPID selection. However, denaturing washes are added as an additional step to remove non-covalent peptide binders to PADI4. The translation incorporates FAO (**2**) and chloroacetylated-D-tyrosine to promote covalent binding and cyclisation, respectively.

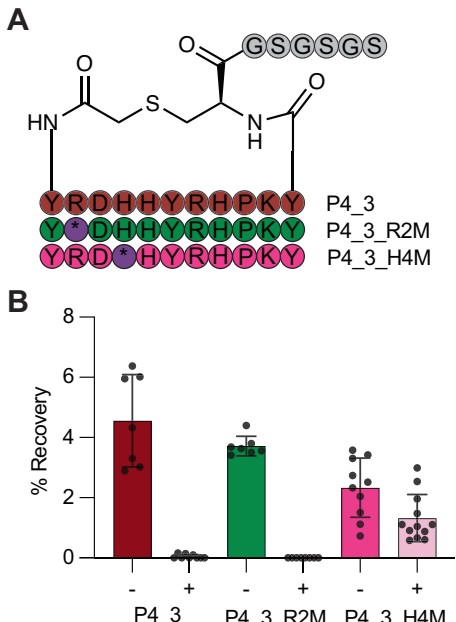

**Fig. 2 | Proof of principle with PADI4 inhibitor PADI4_3. A** Sequences of the different PADI4_3 analogues (abbreviated P4_3) synthesised as mRNA templates and translated, where internal 'M' codons of R2M and H4M are reprogrammed to FAO warhead **2** (*). **B** Clone assay results against PADI4 without (−) or with (+) guanidinium washes. Data shows mean percentage recovery from at least 6 replicates and error bars represent ±1 standard deviation.

$k_{inact}/K_I$ in the order of $10^6$ M$^{-1}$ min$^{-1}$. The most potent of these is >45-fold selective over the other active PADIs. Our current method uses a cysteine reactive warhead, but it is a generally applicable strategy where any weakly reactive covalent warhead can conceivably be incorporated.

## Results and Discussion
### Development of a covalent RaPID platform

Building on previous advances in the development of covalent peptides using encoded libraries[24,27–29], we aimed to develop a covalent peptide discovery platform using RaPID in which an unmasked electrophile would be directly incorporated into the in vitro translated peptides. This first required successful ribosomal incorporation of an electrophile warhead into mRNA-displayed cyclic peptide libraries. We synthesised an unnatural amino acid version of **1**, N-δ-fluoroacetimidoyl ornithine (FAO, **2**, Fig. 1A). To allow flexizyme recognition for aminoacylation onto tRNA we activated the carboxylic acid as the 4-chlorobenzyl thioester (CBT) (FAO-CBT, **3**, Fig. 1A)[33]. This thioester was used because the more commonly used dinitrobenzyl ester of FAO was synthetically intractable. After confirming successful flexizyme-mediated loading onto a short tRNA mimic (Fig. 1B), we tested for ribosomal compatibility. An elongator tRNA with the methionine anticodon (CAU) was aminoacylated with FAO. This aminoacylated tRNA was used to perform in vitro translation using the PURExpress® translation system with methionine omitted. The initiator methionine was reprogrammed to chloroacetyl-D-tyrosine. We translated a peptide template containing a single methionine in the elongator region. MALDI-TOF spectroscopy showed that ribosomal incorporation of FAO was successful (Fig. S1A). As our initial translation efficiency was low, we optimised this by screening four different elongator tRNAs containing variable tRNA T-stems (Fig. S1B)[50]. As the T-stem number increases (1–4) this increases the affinity of the tRNA for elongation factor thermo unstable (EF-Tu), which we observed to correlate with enhanced ribosomal FAO incorporation efficiency. Optimal translation was observed with T-stem 4 which we went on to use in all subsequent experiments. Using this T-stem, similar yields of peptide were achieved as with the control translation suggesting the electrophile-charged tRNA was stable and not interfering with

the activity of components in the translation system (Figure S1B). Additionally, no warhead hydrolysis or adduct formation was observed by MALDI-TOF spectroscopy of the translated peptide (Fig. S1A). Together this confirmed that, at least with this electrophile, covalency can be encoded for in mRNA display with genetic code reprogramming, without the need for masking and post-translational modification.

Denaturing guanidium chloride washes have been used by us and others during the affinity panning step of peptide selections to remove non-covalent binders and only retain covalently binding peptides (Fig. 1C)[24,25,27]. We confirmed that at concentrations up to 8 M, this did not disrupt the interaction between biotinylated PADI4 and streptavidin beads (Figure S2). Based on previous successful selections we chose a concentration of 5 M for the washes[25].

Before performing a full selection we wanted to test our approach using a model protein-peptide target pair. PADI4_3 is a cyclic peptide inhibitor of PADI4 that was recently discovered using a RaPID screen (Fig. 2A)[51]. A cryo-electron microscopy structure of this peptide bound to PADI4 revealed that His4 of PADI4_3 bound in the active site of PADI4, in the position normally occupied by the arginine side chain of substrate peptides (PDB ID: 8R8U). We hypothesised that substitution of this residue with **2** would enable covalent inhibition of PADI4, as the warhead should be positioned to react with the PADI4 active site cysteine, Cys645. To confirm this, PADI4_3_H4(**2**) was synthesised. The linear sequence was synthesised by solid-phase peptide synthesis (SPPS) with an ornithine residue in position 4 of the peptide. Initial attempts to selectively deprotect the ornithine side chain using hydrazine to enable warhead addition also resulted in partial removal of the N-terminal Fmoc group. This led to a mixture of peptide products containing both one and two warheads when ethyl 2-fluoroethanimidate was added. So instead, we devised a strategy where the peptide N-terminus was first deprotected, reacted with N-chloroacetoxysuccinimide, and cyclised with the appropriate cysteine thiol, which was protected with Mmt during the synthesis to allow for selective deprotection. Subsequently, the ornithine side chain was selectively deprotected and reacted with ethyl 2-fluoroethanimidate, before full peptide deprotection, resin cleavage and purification. With the purified peptide in hand, 10 equivalents PADI4_3_H4(**2**) were incubated with PADI4, and intact mass spectrometry (MS) performed. This confirmed that PADI4_3_H4(**2**) covalently bound to PADI4 at a single site (Fig. S3). To evaluate whether the peptides were reacting at the active site cysteine, Cys645, we produced an inactive PADI4 variant in which the Cys645 was substituted with alanine, PADI4 C645A, and performed the same experiment (Fig. S4A–C, S5). No covalent binding between the peptide and PADI4 C645A was observed by intact-MS, confirming that PADI4_3_H4(**2**) was reacting exclusively with the active site Cys645.

Having confirmed that PADI4_3_H4(**2**) was a covalent binder of PADI4, we synthesised three model mRNA templates for use in a test selection. The first template encoded for the wildtype PADI4_3 sequence. The second encoded a sequence where His4 in the PADI4_3 sequence was substituted for a Met codon that could be reprogrammed to FAO, PADI4_3_H4M. The third template encoded a control sequence where an arginine residue in the sequence, Arg2, was replaced by a Met codon, PADI4_3_R2M, which we anticipated would not be correctly positioned to covalently react with PADI4 (Fig. 2A, Fig. S6). We performed a single cycle of RaPID screening (clone assay), with each of the individual mRNA templates, to assess peptide binding. In each case, translated RaPID peptide was incubated with PADI4 at room temperature for 1 hour to allow time for covalent reaction, before affinity panning was performed both with and without denaturing guanidinium chloride washes. qPCR was used to quantify DNA recovery for each peptide. In the absence of guanidinium chloride washes, all three peptides bound to PADI4, whilst only PADI4_3_H4M was retained after guanidinium chloride washes (Fig. 2B). This confirmed that the translated warhead was competent to react with cysteine residues in the target protein, that the guanidinium chloride washes were effective at retaining only covalently bound peptides and that we were not observing high levels of non-specific peptide reaction.

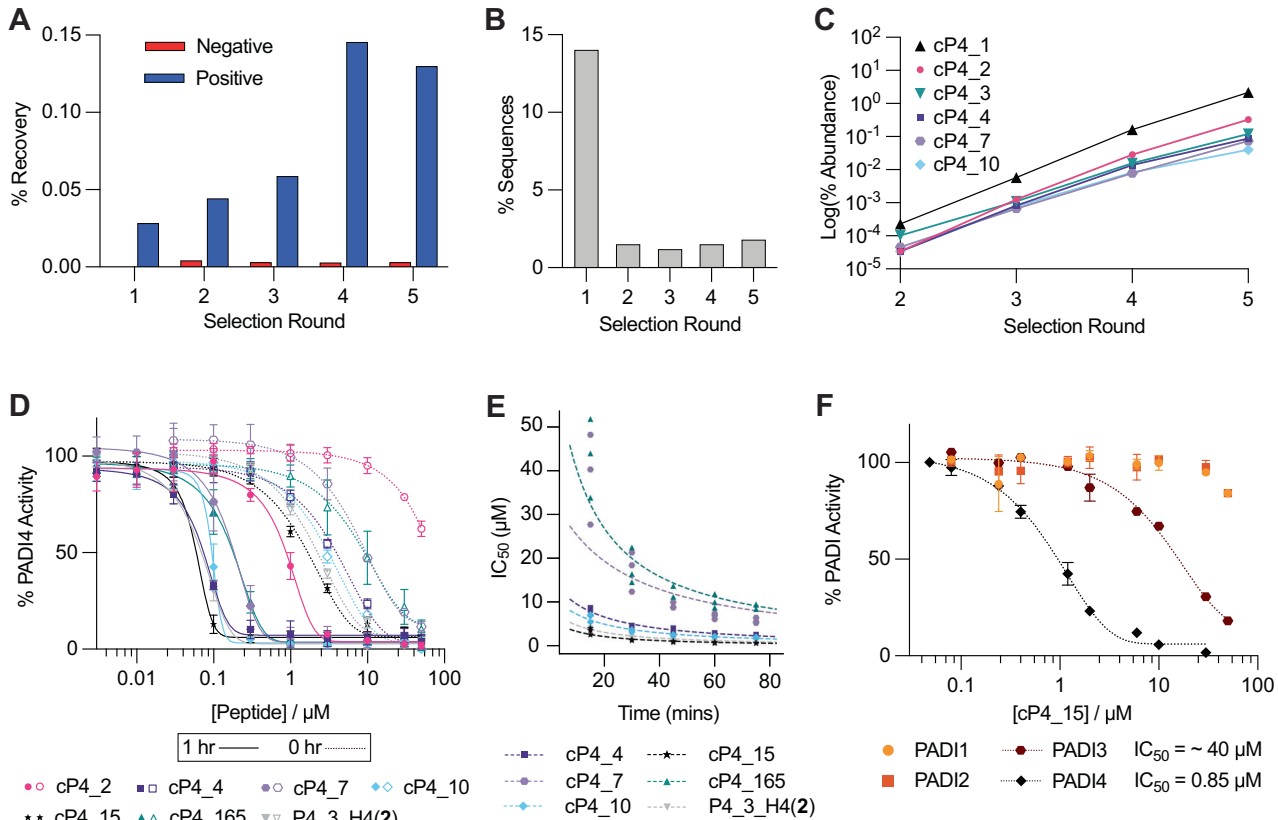

**Fig. 3 | Covalent RaPID selection against PADI4 and peptide characterisation.**
**A** DNA recovery from qPCR after each round of selection from biotinylated beads (negative) and PADI4 beads (positive), compared to the input DNA from each round. **B** Enrichment in warhead-containing peptides. Percentage of sequences from each round of selection which do not contain an internal methionine residue, which encodes the FAO warhead. **C** Enrichment of 6 key sequences over the five rounds of selection. **D** Inhibition COLDER assays with PADI4 and 6 peptides identified in the selection or PADI4_3_H4(**2**). COLDER assays were performed at different peptide concentration (50–0.003 μM) in the presence of 10 mM CaCl₂ either with (1 h) or without (0 h) preincubation of peptide and PADI4. Data is normalised to activity of PADI4 in the presence of 0.1% DMSO. Data shows

mean ± SEM of two independent replicates. Each replicate was done in triplicate. **E** COLDER assays to determine $K_I$ and $k_{inact}$. Apparent IC₅₀ values were determined at 15-minute intervals from three independent replicates and the Krippendorff equation was fitted. **F** COLDER assays to determine selectivity of cP4_15 for PADI4 over PADIs 1-3. Apparent IC₅₀ curves were determined at 15-minute intervals with each of PADIs 1-4. Data for the 60-minute time point is shown. Other time points for PADI3 are shown in Fig. S11A. COLDER assays were performed at different peptide concentration (50–0.08 μM) in the presence of 10 mM CaCl₂ without preincubation with PADI protein. Data is normalised to activity of each PADI in the presence of 0.1% DMSO. Data shows mean ± SEM of at least two independent replicates.

## Covalent RaPID screen of PADI4

Having confirmed that our encoded electrophile RaPID setup could identify covalent cyclic peptides, we set out to perform a de novo peptide screen for covalent binders of PADI4. For this we used an mRNA-displayed library encoding peptides with between six and ten randomised positions, flanked by an initiator codon and a CGSGSGS C-terminal linker. Peptide cyclisation was enabled by flexizyme-mediated reprogramming of the initiator codon to N-chloroacetyl-D-tyrosine which would spontaneously cyclise with the cysteine in the C-terminal linker. All internal Met codons were reprogrammed to **2**. Following translation and reverse transcription, this library was used in a covalent RaPID selection. In each round, the library was preincubated with PADI4 at room temperature prior to denaturing affinity panning against PADI4. We saw low recovery when the library was incubated with biotinylated streptavidin beads (negative selection) and increasing positive library recovery for the first 4 rounds with PADI4-bound beads (Fig. 3A). Following 5 rounds of selection, next-generation sequencing was performed on the DNA libraries recovered after each selection round (Supplementary Data 1). Sequencing results suggested that our RaPID setup rapidly enriched for covalent peptides, because from round 2 onwards 99% of sequences had an internal Met codon, indicating warhead presence (Fig. 3B). Interestingly, however, the sequencing data from later rounds did not resemble typical successful RaPID selections. Although our total library recovery increased through the rounds, rather than finding a smaller number of highly enriched

peptide sequences, contributing substantially to the total recovered libraries, we saw many different individual sequences each with relatively low abundance (Fig. S7). This suggested that reaction with **2** was permissible within a wide range of peptide sequence contexts. Despite this, multiple sequence alignments indicated the enrichment of certain families and a clear increase in the abundance of certain sequences within these families round by round, indicative of target binding (Fig. 3C). We selected 6 of these sequences for synthesis by SPPS and further characterisation.

## Covalent cyclic peptides are potent and selective inhibitors of PADI4

Initially, the synthetic peptides were incubated with PADI4 and PADI4 C645A and samples analysed by intact-MS. This confirmed all peptides were binding at Cys645, in the active site of PADI4, without any additional sites of reaction (Fig. S8). Next, to determine their potency, IC₅₀ values were determined using an established PADI4 activity assay, the Colour Developing Reagent (COLDER) assay, using N-α-benzoyl-ʟ-arginine ethyl ester (BAEE) as the substrate[52]. All peptides showed inhibitory activity against PADI4, both with and without one-hour of preincubation between PADI4 and peptide prior to initiating the assay through addition of BAEE (Fig. 3D). With 1 h preincubation, cP4_15 was the most potent peptide with an IC₅₀ of 52 nM, a slight improvement over PADI4_3_H4(**2**) (IC₅₀ = 61 nM). The least potent peptide, cP4_2, had an IC₅₀ of 870 nM. The

**Table 1 | Sequences of peptides synthesised after the first selection, and the rationally designed PADI4_3_H4(2)**

| Peptide Name | Sequence | IC$_{50}$ Values (µM) | |
| --- | --- | --- | --- |
| | | 0 h preincubation | 1 h preincubation |
| cP4_2 | yIWGL(**2**)D(**2**)SCG | >50 | 0.87 ± 0.06 |
| cP4_4 | ySKYD(**2**)RSPRDCG | 4.4 ± 0.07 | 0.070 ± 0.004 |
| cP4_7 | yVYS(**2**)KEWKYCG | 8.0 ± 1.2 | 0.16 ± 0.02 |
| cP4_10 | yWY(**2**)NWDFNKRCG | 3.1 ± 0.01 | 0.093 ± 0.002 |
| cP4_15 | yLD(**2**)HYSSKLYCG | 1.6 ± 0.03 | 0.052 ± 0.002 |
| cP4_165 | yVY(**2**)DCEWINRAG | 11.8 ± 4.9 | 0.17 ± 0.01 |
| PADI4_3_H4(**2**) | yRD(**2**)HYRHPKYCG | 2.0 ± 0.01 | 0.065 ± 0.007 |

Where (**2**) is the FAO warhead and y is chloroacetyl-D-tyrosine which is cyclised with the cysteine residue in each peptide. Their corresponding IC$_{50}$ values from COLDER assay with or without 1 h preincubation of peptide and PADI4 are also shown. Data shows mean ± SEM of two independent replicates.

**Table 2 | Kinetic parameters of peptides. Both COLDER and SPR values show mean ± SEM from three independent replicates**

| | SPR | | | Colders | | |
| --- | --- | --- | --- | --- | --- | --- |
| | $K_i$ (µM) | $k_{inact}$ (min$^{-1}$) | $k_{inact}/K_i$ (M$^{-1}$ min$^{-1}$) | $K_I$ (µM) | $k_{inact}$ (min$^{-1}$) | $k_{inact}/K_I$ (M$^{-1}$ min$^{-1}$) |
| cP4_2 | 1.3 ± 0.5 | 0.12 ± 0.1 | 92,000 | - | - | - |
| cP4_4 | 0.12 ± 0.01 | 0.11 ± 0.1 | 909,000 | 1.9 ± 0.1 | 0.13 ± 0.01 | 74,000 |
| cP4_7 | 1.6 ± 0.1 | 0.55 ± 0.02 | 343,000 | 4.3 ± 1.1 | 0.077 ± 0.052 | 18,000 |
| cP4_10 | 0.12 ± 0.1 | 0.23 ± 0.01 | 2,007,000 | 1.5 ± 0.3 | 0.13 ± 0.03 | 87,000 |
| cP4_15 | 0.16 ± 0.03 | 0.25 ± 0.04 | 1,594,000 | 0.78 ± 0.06 | 0.17 ± 0.02 | 213,000 |
| cP4_165 | 0.62 ± 0.06 | 0.066 ± 0.007 | 107,000 | 8.9 ± 1.1 | 0.16 ± 0.01 | 17,000 |
| PADI4_3 H4(2) | 0.015 ± 0.002 | 0.061 ± 0.011 | 4,083,000 | 1.1 ± 0.2 | 0.16 ± 0.03 | 149,000 |

same trend in IC$_{50}$ values was observed without preincubation of the peptides with PADI4, however, the IC$_{50}$ values were much higher (Table 1). This time-dependent improvement in IC$_{50}$ is indicative of covalent inhibition. Therefore, to characterise the covalent behaviour further, kinetic parameters were determined using a modified COLDER assay design. Varied concentrations of each peptide were incubated with PADI4 and BAEE and the reactions were quenched at 15-minute time intervals. Apparent IC$_{50}$ values were calculated at each time point. This allowed an IC$_{50}$ vs time correlation to be determined and fitted to the Krippendorff Equation which allows determination of $K_I$ and $k_{inact}$ values (Fig. 3E, Table 2)[53,54]. The $k_{inact}/K_I$ values determined were up to 10-fold higher than those previously reported for PADI4 covalent inhibitors[49].

Peptide binding to PADI4 was also characterised by surface plasmon resonance (SPR) (Fig. S9). Consistent with the covalent mode of action seen by intact MS and activity assays, during our single-cycle kinetics experiments, the baseline of the SPR binding curves did not return to zero, despite elongated dissociation times, indicative of covalent binding. Consequently, a simple 1:1 binding model did not describe the data well. By fitting the data using a two-state reaction model, in which the rate constant for the reverse second step ($k_{-2}$) was set to zero, the equilibrium constant of the reversible binding step, $K_i$, could be calculated as the ratio of $k_{-1}/k_{+1}$ and the rate constant of the irreversible chemical step, $k_{inact}$, as $k_{+2}$ (Table 2). In most cases, the $k_{inact}$ values are similar whilst the $K_I$ values from the COLDERs are generally an order of magnitude higher. Despite this, the rank order of peptides by $k_{inact}/K_i$ is similar. Differences are only observed between the three most potent peptides, which is where we anticipate the most error in our fitting for both methods.

To test the specificity of the FAO warhead-containing peptides, PADIs 1–3 were expressed and confirmed to be active in citrullination assays (Fig. S10). The binding of the most potent inhibitor, cP4_15, was tested using COLDER assays with PADIs 1–3 (Fig. 3F). These experiments indicated that cP4_15 was highly selective for PADI4. Approximately 45-fold selectivity was observed over PADI3. Even greater selectivity was observed

over PADI1 and PADI2, with very minimal inhibition seen even at the highest concentration of peptide tested (50 µM). $k_{inact}/K_i$ values for PADI3 were indeterminable due to weak inhibition causing incomplete IC$_{50}$ curves using the range of inhibitor concentrations we were able to use in our assay (Figure S11A). Peptide cP4_4 was also tested and showed 8-fold selectivity for PADI4 over PADI3 (Fig. S11B–D). As with cP4_15, very minimal inhibition of PADI1 and PADI2 was observed. These results indicate that RaPID selections can be used to find highly selective covalent inhibitors of PADI4 and that sequence context of the FAO warhead strongly affects its selectivity.

To further understand the contribution of **2** to PADI4 binding and inhibition, variants of cP4_4 where **2** was substituted for arginine (Arg4) or citrulline (Cit4) were synthesised and tested (Fig. 4). Surprisingly, neither peptide showed any inhibition of PADI4 activity as measured using the COLDER assays (Table 3, Fig. 4B). Arg4 did, however, bind reversibly to PADI4 with an affinity of 2.0 µM, as measured by SPR, whilst Cit4 showed negligible binding at the concentrations tested (Fig. S10). This is consistent with Arg4 acting as a substrate of PADI4; on binding to PADI4 in the COLDER assays it is converted to Cit4 which no longer binds and hence inhibition is not observed. To assess whether warhead **2** was essential for inhibition, we additionally decided to synthesise the H-amidine analogue, Me4. Me4 had a comparable affinity to Arg4, however unlike Arg4 it also acted as a weak inhibitor of PADI4, consistent with our hypothesis that lack of PADI4 inhibition by Arg4 is due to it being turned over as a substrate (Fig. 4B). The reduction in affinity and PADI4 inhibition of Me4 relative to the F-amidine parent cP4_4 suggests that the fluorine atom forms important interactions within the active site of PADI4 that are crucial for binding, as well as acting as the leaving group.

Finally, we made the Cl-amidine analogue (Cl4) to see what effect this more reactive electrophile would have on the potency of the peptide. The IC$_{50}$ values were similar to those of cP4_4, although without preincubation, the IC$_{50}$ was slightly higher (Table 3). Interestingly, when we determined $K_i$ and $k_{inact}$ values using SPR, Cl4 had a weaker $K_i$ but higher $k_{inact}$ (Fig. S12,

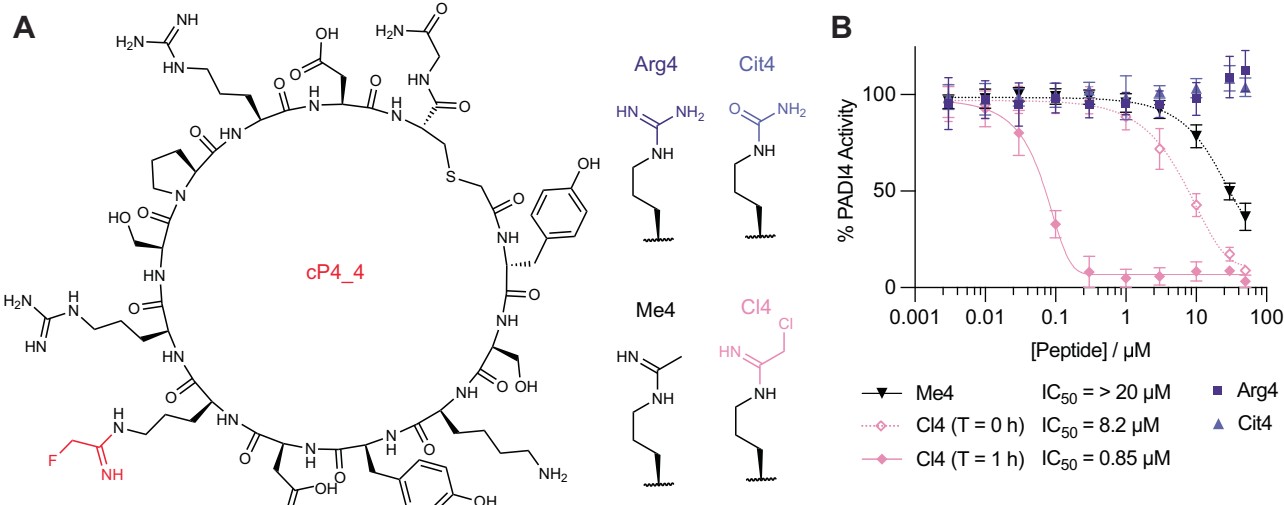

**Fig. 4 | Testing the inhibitory effect of the cP4_4 peptide variants on PADI4.**
**A** The displayed structure of cP4_4 with FAO highlighted in red, and the four modifications made at the FAO position. **B** Inhibition COLDER assays with PADI4 and cP4_4 variant peptides where FAO was substituted with variable groups. COLDER assays were performed at different peptide concentration (50–0.003 μM) in the presence of 10 mM $CaCl_2$. For the reversible warheads, no preincubation between peptide and PADI4 was performed. For Cl4 data is shown either with (1 h) or without (0 h) preincubation of peptide and PADI4. Data is normalised to activity of PADI4 in the presence of 0.1% DMSO. Data shows mean ± SEM of at least two independent replicates.

**Table 3 | Summary of binding affinities and in vitro activity of cP4_4 and its variants where the fluoroacetimidoyl ornithine (FAO) warhead was replaced by arginine (Arg4), citrulline (Cit4), acetimidoyl ornithine (Me4) or chloroacetimidoyl ornithine (Cl4)**

| | SPR | | | | COLDERS | |
|---|---|---|---|---|---|---|
| | $K_i$ (μM) | $k_{inact}$ (min$^{-1}$) | $k_{inact}/K_i$ (M$^{-1}$ min$^{-1}$) | $K_D$ (μM) | IC$_{50}$ (μM) T = 0 h | IC$_{50}$ (μM) T = 1 h |
| Arg4 | – | – | – | 2.0 ± 0.2 | >100 | |
| Cit4 | – | – | – | >10 | >100 | |
| Me4 | – | – | – | 1.7 ± 0.4 | >20 | |
| Cl4 | 4.0 ± 0.2 | 0.28 ± 0.02 | 70,000 | – | 8.2 ± 0.8 | 0.066 ± 0.009 |
| cP4_4 | 0.12 ± 0.01 | 0.11 ± 0.003 | 909,000 | – | 4.4 ± 0.07 | 0.070 ± 0.004 |

Surface plasmon resonance (SPR) data shows mean ± SEM of three independent replicates. COLDER data shows mean ± SEM of at least two independent replicates.

Table 3). This is consistent with the larger chlorine atom sterically hindering binding, but increasing the rate of the covalent reaction step[46]. This warhead was also confirmed to be more reactive by intact MS, which showed that Cl4 could covalently react twice with PADI4, once at the active site C645 and a second time at an unknown location (Fig. S13).

**Expansion of covalent RaPID selections**
Although several potent covalent inhibitors had been found, we decided to test whether we could further optimise the selection conditions with the hope of promoting greater discrimination between more and less potent inhibitors. To this end, we repeated the selection starting from round 2, reducing incubation of peptides with PADI4 to only 15 minutes at 0 °C. As we expected, we saw a reduction in positive library recovery which matched the increased stringency, but recovery still increased round-by-round (Fig. 5A). After sequencing the recovered libraries, we again observed that the sequences from round 2 onward had a low percentage of sequences which did not encode for a warhead (Fig. 5B, Supplementary Data 2). The sequencing results showed the most enriched peptides were those found originally (cP4_4, cP4_7 and cP4_10), but with greater enrichment (Fig. 5C). We saw an abolishment from the selection of the poor hit cP4_2 and the peptide hit without the warhead cP4_3. Two further peptides that were uniquely identified in this second screen, cP4_13 and cP4_18, were synthesised. Both were shown to covalently bind to C645 of PADI4 by intact MS (Fig. S14). cP4_13 closely resembled cP4_2 from the first selection, but with only one warhead **2** present. Neither peptide had strong inhibitory

activity against PADI4 (Fig. S15) and cP4_13 was considerably less active than cP4_2. Characterisation by SPR also showed that these peptides were among some of the poorest binders synthesised (Fig. S16). These results showed that the alternative selection conditions did not eradicate the least potent hits, however their enrichment levels were lower relative to the more active cP4_4, cP4_7 and cP4_10, which might have helped with initial peptide selection for synthesis by SPPS.

Given all our identified peptides bound to the active site cysteine of PADI4, in parallel we performed a selection on biotinylated PADI4 C645A (Fig. S17), to see if we could promote identification of peptide binders at alternative cysteine residues in PADI4 when the favoured C645 was not available. We chose the same selection conditions as the initial selection to increase our probability of identifying even poor binders. However, there was very minimal enrichment of positive recovery with the PADI4 C645A-bound beads, which was always far exceeded by the negative recovery with biotinylated streptavidin beads (Fig. 5D). Nonetheless, we sequenced the recovered libraries. This confirmed that there was no round-by-round enrichment in warhead-containing sequences (Fig. 5E, Supplementary Data 3). Of the top 30 most enriched sequences, few contained warhead **2**. Those that did, were also found in previous selections and did not significantly increase in abundance over the course of the selection, suggesting that they are contaminations that bind C645 (Fig. 5F). The other sequences most frequently enriched had lost the CGSGSGS linker or contained a very large proportion of Cys residues. There was also a low number of sequences for the later rounds (Figure S18). These factors all indicated that even

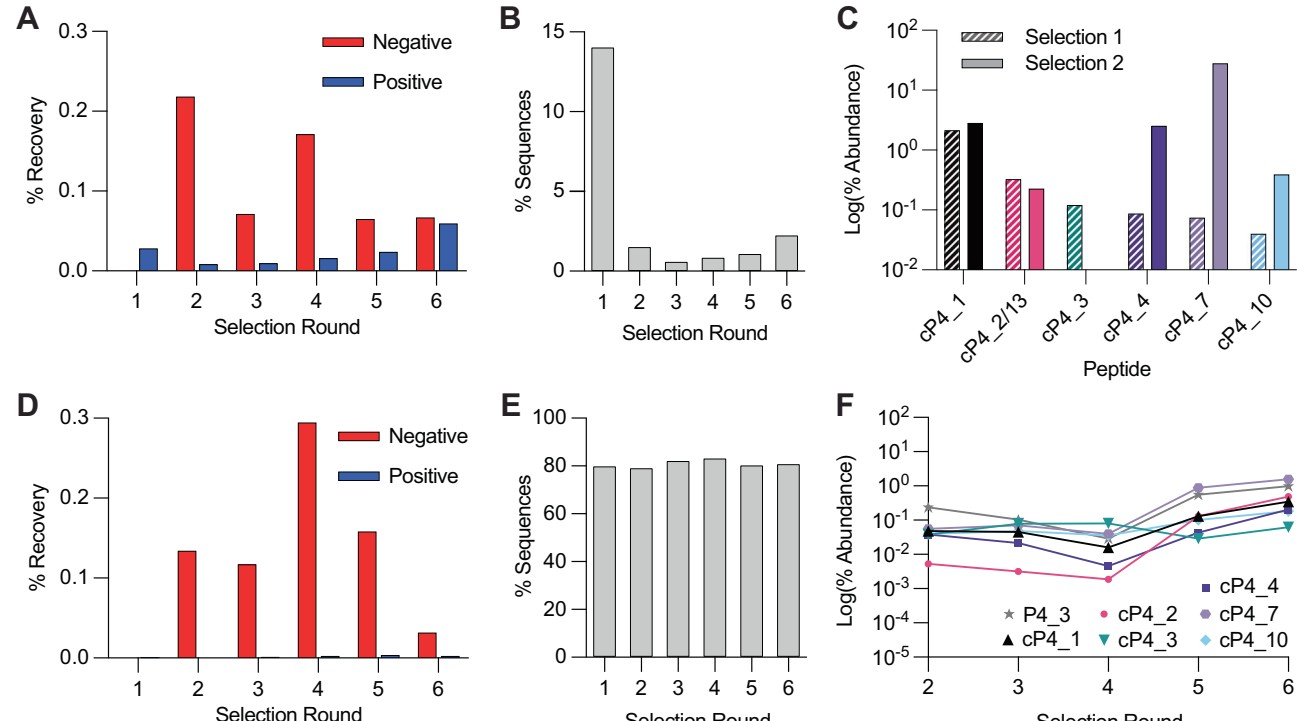

**Fig. 5 | Covalent RaPID selection against PADI4 with increased stringency and against PADI4 C645A. A** and **D** DNA recovery from qPCR after each round of selection against PADI4 and PADI4 C645A, respectively, from biotinylated beads (negative) and PADI4 beads (positive), compared to the input DNA from each round. **B** and **E** Enrichment in warhead-containing peptides for PADI4 and PADI4 C645A selections, respectively. Percentage of sequences from each round of selection which do not contain an internal methionine residue, which encodes the FAO warhead. **C** Enrichment of key sequences at the 5th round of selection from selection one (hashed bars) and selection two (filled bars), where cP4_13 from selection two has high sequence homology with cP4_2 from selection one. **F** Low levels of enrichment of key sequences over the six rounds of selection against PADI4 C645A.

through presentation on a tight binding cyclic peptide, warhead **2** could not be forced to react at alternative Cys residues in PADI4, hence **2** requires a specifically activated, nucleophilic Cys, like the active site C645.

## Conclusions

In summary, we show here the development of a RaPID workflow that can select for covalent reaction between peptide and target by directly incorporating an electrophilic warhead into each member of the peptide library, whilst negatively selecting for non-covalent interactions, even if they are low nanomolar binders, like PADI4_3. We have used flexizymes to genetically encode covalency into mRNA display, specifically **2**, a cysteine reactive, fluoroamidine-based warhead. Using this approach we have developed some of the most potent and selective covalent inhibitors of PADI4 identified to date. In the future, these covalent libraries could be used to identify covalent cyclic peptide inhibitors of a range of related therapeutically relevant enzymes, including bacterial arginine deiminases, the cardiovascular target, DDAH, and the inflammatory target, STING, which has previously been shown to react with Cl-amidine[55–57]. More generally, we also envisage this RaPID workflow could be applied with any other covalent warhead which can be loaded using flexizymes to target a wider range of cysteine and non-cysteine residues in therapeutic targets.

## Methods
### Peptide synthesis
Peptide synthesis was performed by solid phase peptide synthesis using a Gyros Protein Technologies PreludeX automated synthesizer (Gyros Protein Technologies AB, Sweden). The purity and masses of all peptides was determined using analytical HPLC (Fig. S19). Aside from citrulline and D-tyrosine, which were commercially available, unnatural amino acids were made from ornithine residues which were incorporated within the peptide sequence, orthogonally deprotected and reacted with ethyl

2-fluoroethanimidate hydrochloride. In the case of Me4, ornithine was reacted with 2,2,2-trichloroethyl acetimidate hydrochloride and for Cl4 synthesis, ethyl 2-chloroethanimidate hydrochloride was used. All NMR spectra are provided in Supplementary Data 4. Detailed methods are provided in Supplementary Method S7.

### Covalent RaPID
In vitro selections were performed against bio-His-PADI4 and bio-His-PADI4 C645A following previously described protocols. Briefly, initial DNA libraries (including 6-10 degenerate NNK codons in a ratio 0.0018 $NNK_{n=6}$:0.032 $NNK_{n=7}$:1 $NNK_{n=8}$:32 $NNK_{n=9}$:80 $NNK_{n=10}$) (see Table S1 for DNA sequence) were transcribed to mRNA using T7 RNA polymerase (37 °C, 16 h) and ligated to Pu_linker (Table S1) using T4 RNA ligase (30 min, 25 °C). First round translations were performed on a 75 μL scale, with subsequent rounds performed on a 5 μL scale. Translations were carried out (1 h, 37 °C then 12 min, 25 °C) using a custom methionine(-) Flexible In vitro Translation system containing additional ClAc-D-Tyr-tRNA$^{fMet}_{CAU}$ (25 μM) and N-δ-Fluoroacetimidoyl ornithine-CBT (**3**, 25 μM, Figs. S23 and S24). Ribosomes were then dissociated by addition of EDTA (18 mM final concentration, pH 8) and library mRNA reverse transcribed using MMLV RTase, Rnase H Minus (Promega). Reaction mixtures were buffer exchanged into selection buffer (50 mM HEPES, pH 7.5, 150 mM NaCl, 2 mM DTT, 10 mM CaCl₂) using 1 mL homemade columns containing pre-equilibrated Sephadex resin (Cytiva). Blocking buffer was added (1 mg/mL sheared salmon sperm DNA (Invitrogen), 0.1% acetyl-BSA final (Invitrogen)). Libraries were incubated with negative selection beads (3×30 min, 4 °C). Libraries were then incubated with bead-immobilised bio-His-PADI4 or bio-His-PADI4 C645A (200 nM, rt for 1 h or 4 °C for 15 min) before washing (3 ×1 bead volume selection buffer, 4 °C then 3 ×1 bead volume 5 M guanidinium HCl, 4 °C) and elution of retained mRNA/DNA/peptide hybrids in PCR buffer (95 °C, 5 min). Library

recovery was assessed by quantitative real-time PCR relative to a library standard, negative selection and the input DNA library. Recovered library DNA was used as the input library for the subsequent round. Following completion of the selections, double indexed libraries (Nextera XT indices) were prepared and sequenced on a MiSeq platform (Illumina) using a v3 chip as single 151 cycle reads. Sequences were ranked by total read numbers and converted into their corresponding peptides sequences for subsequent analysis (Supplementary Files 1–3).

## Bead preparation

For PADI4 immobilisation, bio-His-PADI4 or bio-His-PADI4_C645A were incubated with magnetic streptavidin beads (Invitrogen) (4 °C, 15 min to an immobilisation level of 0.9 pmol/µL beads) immediately before use in the selection. Biotin was added to cap unreacted streptavidin sites (25 µM final, 4 °C, 15 min). Beads were washed 3 ×1 bead volume selection buffer and left on ice for use in the selection. Negative beads were prepared similarly except that only selection buffer or selection buffer plus biotin (25 µM) were added to beads and following washing these two variants were mixed.

## COLDER assays

PADI4 citrullination activity was analysed using the COLDER assay in 96-well plates[52]. Peptide dilutions were prepared from a 500 µM stock, to give a 10 times concentrated dilution series (500 µM, 300 µM, 100 µM, 30 µM, 10 µM, 3 µM, 1 µM, 0.3 µM, 0.1 µM and 0.03 µM) in COLDER buffer (50 mM HEPES, 150 mM NaCl and 2 mM DTT, pH 7.5) containing 1% DMSO. In triplicate, each was diluted 10-fold further when mixed with 50 nM His-PADI4, 0.6 mg/mL BSA and 10 mM CaCl$_2$, in COLDER buffer. With or without one hour of incubation, 10 mM $N^\alpha$-Benzoyl-L-arginine ethyl ester hydrochloride (BAEE, Merck) was added to initiate the reaction (50 µL final volume). After 30 min at rt, EDTA (50 mM final concentration) was used to quench the reaction and 200 µL of COLDER solution containing 20 mM Diacetyl monoxime/2,3-butanedione monoxime (Merck), 0.5 mM Thiosemicarbazide (Acros Organics), 2.25 M H$_3$PO$_4$, 4.5 M H$_2$SO$_4$ and 1.5 mM NH$_4$Fe(SO$_4$)$_2$.12H$_2$O was added to each well. Samples were incubated for 20 min at 95 °C before measuring absorbance at 540 nm on a CLARIOstar Plus (BMG LABTECH). Data analysis was performed with GraphPad Prism. Data are presented as the average ± standard error of the mean from at least two independent replicates.

## Incubation time-dependent potency IC$_{50}(t)$

To determine $K_I$ and $k_{inact}$ of covalent peptide inhibitors, COLDER assays were used[52]. Peptide dilutions were prepared using 5-fold dilutions from 500 µM and 300 µM to 0.8 µM and 2.4 µM, respectively, at 1% DMSO in COLDER buffer (50 mM HEPES, 150 mM NaCl and 2 mM DTT, pH 7.5). The 9 peptide dilutions were added to a 96-well plate, alongside a 1% DMSO control. An equal volume of 10 mM BAEE was added to each well and the solution was homogenised by pipette mixing. This was mixed with 50 nM His-PADI4, 0.6 mg/mL BSA and 10 mM CaCl$_2$, in COLDER buffer, to bring the final volume to 300 µL in each well. Alternatively, His-PAD1 (100 nM) /His-PADI2 (100 nM) /His-PADI3 (250 nM) were used. For assays with PADI1 and PADI3, 10 mM $N^\alpha$-benzoyl-L-arginine methyl ether (BAME) was used as the substrate instead of BAEE[48,58,59]. At 15-minute intervals, for 5 timepoints, 50 µL of solution was taken from each well and quenched with 10 µL EDTA (300 mM). 200 µL of COLDER solution containing 20 mM diacetyl monoxime/2,3-butanedione monoxime (Merck), 0.5 mM thiosemicarbazide (Acros Organics), 2.25 M H$_3$PO$_4$, 4.5 M H$_2$SO$_4$ and 1.5 mM NH$_4$Fe(SO$_4$)$_2$.12H$_2$O was added to each well. Samples were incubated for 20 min at 95 °C before measuring absorbance at 540 nm on a CLARIOstar Plus (BMG LABTECH). IC$_{50}$ values for each time point were determined using non-linear regression with GraphPad Prism. Incubation time–dependent potency IC$_{50}(t)$ against incubation time was fitted to the Krippendorff equation (below) to determine $K_I$ and $k_{inact}$ using a Python script[53,59]. $K_m$ values of 1.36 mM and 10.8 mM were used for PADI4 (with BAEE) and PADI3 (with BAME), respectively, based on published data[48,59].

$$\mathrm{IC}_{50}(t) = K_I \left(1 + \frac{S}{K_M}\right) \cdot \left(\frac{2 - 2e^{-\eta_{\mathrm{IC}_{50}} \cdot k_{inact} \cdot t}}{\eta_{\mathrm{IC}_{50}} \cdot k_{inact} \cdot t} - 1\right)$$

$$\text{Where} \quad \eta_{\mathrm{IC}_{50}} = \frac{\mathrm{IC}_{50}(t)}{K_I \left(1 + \frac{S}{K_M}\right) + \mathrm{IC}_{50}(t)}$$

## Reporting summary

Further information on research design is available in the Nature Portfolio Reporting Summary linked to this article.

## Data availability

Detailed Supplementary Methods and Supplementary Figs. are provided in the Supplementary Information. All data supporting the results is available as source data in Supplementary Data 5. NGS Sequencing data has been uploaded in the NIH Short Read Archive (SRA) under the accession number PRJNA1188087 and deconvoluted peptide sequence lists data are provided in Supplementary Data 1-3.

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

## Acknowledgements

This work was funded by UK Research and Innovation (UKRI) under the UK government's Horizon Europe funding guarantee [grant number EP/X020878/1]. This work was also supported by the Francis Crick Institute which receives its core funding from Cancer Research UK (CC2030), the UK Medical Research Council (CC2030), and the Wellcome Trust (CC2030). We would like to thank the Crick Advanced Sequencing Science Technology Platform for assistance with next generation sequencing and Sarah Maslen of the Crick Proteomics Science Technology Platform for performing the intact mass spectrometry for this work. Additional thanks to the Crick Chemical Biology Science Technology Platform for their advice about peptide synthesis, especially Dhira Joshi. Final thanks go to Jack Williams for his help with PADI expression and to Samrah Bourhan for her help during her BSc project. Illustrative figures were created using BioRender.com.

## Author contributions

I.R.M. and E.D.D.C. planned and executed experiments and analysed data; S.K. analysed SPR data; L.J.W. conceptualised the project, obtained funding and supervised the work; I.R.M. and L.J.W. wrote the manuscript with help from all authors.

## Funding

## Competing interests

The authors declare no competing interests.
