## [Peer review file · Communications Chemistry]

Discovering Covalent Cyclic Peptide Inhibitors of Peptidyl Arginine Deiminase 4 (PADI4) Using mRNA-Display with a Genetically Encoded Electrophilic Warhead

Corresponding Author: Dr Louise Walport

Version 0:

Reviewer comments:

Reviewer #1

(Remarks to the Author)

The manuscript by Mathiesen et al. describes the development of a mRNA display screen to identify covalent binding cyclic peptides for the enzyme PADI4. The authors show that a non-natural amino acid containing a fluoroamidine electrophile can be incorporated into peptides by loading onto a tRNA using the flexizyme system. The authors show that the resulting electrophilic peptides do not react with other proteins in the translation mix and selection allows direct identification of potent covalent inhibitors of the target enzyme. They show that compounds bind through direct covalent binding to an active site cysteine.

Overall, this is a very nice study that shows mRNA display can be used to identify covalent binding cyclic peptides by incorporating an electrophilic warhead directly into the translated peptides. While others have also shown this method works, the current study uses a new warhead which is not protected or masked in any way. The strengths of the study are the solid methodology developed, clear presentation of the results and interesting findings regarding the types of sequences selected when using covalent ligand selection. The authors also do very rigorous characterization of the hit peptides to show that the binding is specific and that selection using a mutant form of the target results in non-specific binding sequences. The only minor weakness of this study is that the authors start by using a simple amino acid swap approach in which they change out one residue of a non-covalent binding cyclic peptide that they identified for PADI4 in another study and end up with a potent covalent binding inhibitor that is basically as good as anything they get from selections. This result suggests that maybe selection is not necessary for this target so maybe this is a bad example for this overall approach. In addition, there is no data regarding the selectivity of the final most potent hits. It seems the benefit of doing all the selection using big cyclic peptide scaffolds is that the molecules should bind more selectively to the desired target. As pointed out in the text, fluoroamidines have been shown to inhibit PADI4 but their main weakness is that they are not selective over PADI1 when used without a peptide scaffold. Why were none of the cyclic peptide hits from this study tested against other PADI enzymes to show selectivity? Other than these points, I think this is a nice piece of work that should be published as it adds to the growing toolbox of screening methods to find covalent binding peptide ligands for important targets.

Specific points:

The authors should show data for their compounds against other PADI targets to give an indication of specificity. If there are not good enzyme assays for each enzyme it should be possible to make an alkyne labeled version of the best hit and show that it labels PADI4 over PADI1 when spiked into a total lysate.

Page 5 lines 116-119. It says there was no warhead hydrolysis or reaction with components of the translation system but it is not explained how this was determined and there is no data to support this claim. It would be good to know if they could confirm lack of reactivity of the electrophile with other proteins.

When the authors introduce the PADI4_3 peptide that was published on BioRxiv by their group, they should cite the sequence and structure shown in figure 2A. I was looking for it for a while before I realized it was not shown until later in figure 2A.

The choice of colors for C14 T=0 and T=1h is bad. It is very difficult to see differences in those colors and dotted lines.

Reviewer #2

(Remarks to the Author)

The manuscript titled "Discovering Covalent Cyclic Peptide Inhibitors of Peptidyl Arginine Deiminase 4 (PADI4) Using mRNA Display with a Genetically Encoded Electrophilic Warhead" by the Walport group presents a study on the ribosomal translation of a fluoroamidine warhead that specifically targets cysteine residues. Using FIT system, the authors successfully integrated this warhead into peptide libraries. This approach enabled the de novo discovery of covalent binders and potent inhibitors of PADI4, displaying selective reactivity at the active site cysteine. The findings presented are both novel and significant. The manuscript is well-written, with high-quality characterization and thorough experimentation. I believe that the manuscript is worthy of publication in Communications Chemistry after minor revisions.

Minor concerns:

1. The authors demonstrated the successful ribosomal incorporation of a fluoroamidine warhead in the presence of cysteine, which selectively reacts with the chloroacetyl group without interfering with the warhead. But during the peptide synthesis, the authors initially performed cyclization, and then reacted the cyclized product with ethyl 2-fluoroethanimidate after removing the DDE protecting group to produce macrocyclic peptides containing the fluoroamidine warhead. It would be better to include a more detailed explanation of this process. Is the warhead sensitive to piperidine, or is there another reason?

2. The authors conducted SPR experiments to assess peptide binding to PADI4. The data analysis used a two-state reaction model, with the reverse rate constant for the second step (k_{-2}) set to zero. This allowed the determination of K_i and k_{inact} values, rather than the classical k_{a1} , k_{d1} , k_{a2} , k_{d2} , and K_D values. A more detailed explanation of this analysis would be beneficial. Additionally, if the SPR results support a covalent binding mode, further discussion on this point may enhance the manuscript.

3. The authors might consider including a ChemDraw structure of one or more important macrocyclic peptides containing the warhead to improve readability and comprehension in the main text of the manuscript.

4. Did the authors investigate the selectivity of the selected peptides for PADI4 over PADI1, as mentioned in the introduction?

Version 1:

Reviewer comments:

Reviewer #1

(Remarks to the Author)

The authors have nicely addressed my concerns, specifically by adding in data for the specificity of lead molecules. The only minor issue is that in the new Fig 3F the symbol key does not match the graph. The colors seem correct but the shapes are not matching with the key.

Reviewer #2

(Remarks to the Author)

The authors have adequately addressed my concerns, and I recommend publishing it as is.

REVIEWER COMMENTS

We wish to thank both reviewers for their positive and helpful comments on our manuscript. We have addressed their specific points in the manuscript accordingly. We provide a point-by-point response below. Changes are highlighted in the manuscript.

Reviewer #1 (Remarks to the Author):

The manuscript by Mathiesen et al. describes the development of a mRNA display screen to identify covalent binding cyclic peptides for the enzyme PADI4. The authors show that a non-natural amino acid containing a fluoroamidine electrophile can be incorporated into peptides by loading onto a tRNA using the flexizyme system. The authors show that the resulting electrophilic peptides do not react with other proteins in the translation mix and selection allows direct identification of potent covalent inhibitors of the target enzyme. They show that compounds bind through direct covalent binding to an active site cysteine.

Overall, this is a very nice study that shows mRNA display can be used to identify covalent binding cyclic peptides by incorporating an electrophilic warhead directly into the translated peptides. While others have also shown this method works, the current study uses a new warhead which is not protected or masked in any way. The strengths of the study are the solid methodology developed, clear presentation of the results and interesting findings regarding the types of sequences selected when using covalent ligand selection. The authors also do very rigorous characterization of the hit peptides to show that the binding is specific and that selection using a mutant form of the target results in non-specific binding sequences. The only minor weakness of this study is that the authors start by using a simple amino acid swap approach in which they change out one residue of a non-covalent binding cyclic peptide that they identified for PADI4 in another study and end up with a potent covalent binding inhibitor that is basically as good as anything they get from selections. This result suggests that maybe selection is not necessary for this target so maybe this is a bad example for this overall approach. In addition, there is no data regarding the selectivity of the final most potent hits. It seems the benefit of doing all the selection using big cyclic peptide scaffolds is that the molecules should bind more selectively to the desired target. As

pointed out in the text, fluoroamidines have been shown to inhibit PADI4 but their main weakness is that they are not selective over PADI1 when used without a peptide scaffold. Why were none of the cyclic peptide hits from this study tested against other PADI enzymes to show selectivity? Other than these points, I think this is a nice piece of work that should be published as it adds to the growing toolbox of screening methods to find covalent binding peptide ligands for important targets.

Specific points:

The authors should show data for their compounds against other PADI targets to give an indication of specificity. If there are not good enzyme assays for each enzyme it should be possible to make an alkyne labeled version of the best hit and show that it labels PADI4 over PADI1 when spiked into a total lysate.

We agree that this is important data to include and have tested the selectivity of two of our peptides – the most active cP4_15 and our main test peptide cP4_4 – against the other active PADIs, PADI1/2/3. Changes made are highlighted in yellow in the manuscript and pasted below.

To test the specificity of the FAO warhead-containing peptides, PADIs 1-3 were expressed and confirmed to be active in citrullination assays (Figure S10). The binding of the most potent inhibitor, cP4_15, was tested using COLDER assays with PADIs 1-3 (Figure 3F). These experiments indicated that cP4_15 was highly selective for PADI4. Approximately 45-fold selectivity was observed over PADI3. Even greater selectivity was observed over PADI1 and PADI2, with very minimal inhibition seen even at the highest concentration of peptide tested (30 μ M). k_{inact}/K_i values for PADI3 were indeterminable due to weak inhibition causing incomplete IC₅₀ curves using the range of inhibitor concentrations we were able to use in our assay (Figure S11A). Peptide cP4_4 was also tested and showed 8-fold selectivity for PADI4 over PADI3 (Figure S11B-D). As with cP4_15, very minimal inhibition of PADI1 and PADI2 was observed. These results indicate that RaPID selections can be used to find highly selective covalent inhibitors of PADI4 and that sequence context of the FAO warhead strongly affects its selectivity.

Figure S10. Characterisation of other PADIs. **A** SDS PAGE gel of different quantities of purified PADI1 **B** SDS PAGE gel of different quantities of purified PADI2 **C** SDS PAGE gel of different quantities of PADI3. **D** COLDER assays show that all four PADIs are catalytically active. Individual data points are shown from two repeats each with three replicates and error bars represent the mean ± 1 s.d..

Figure 3. F COLDER assays to determine selectivity of cP4_15 for PADI4 over PADIs 1-3. Apparent IC₅₀ curves were determined at 15-minute intervals with each of PADIs 1-4. Data for the 60-minute time point is shown. Other time points for PADI3 are shown in Figure S11A. COLDER assays were performed at different peptide concentration (50 – 0.08 μM) in the presence of 10 mM CaCl₂ without preincubation with PADI protein. Data is normalised to activity of each PADI in the presence of 0.1% DMSO. Data shows mean ± SEM of at least two independent replicates.

Figure S11. COLDER assays with cP4_4/cP4_15 and PADIs 1-4. **A** and **B** COLDER assays of cP4_15 (**A**) or cP4_4 (**B**) with PADI3. COLDER assays were performed at different peptide concentration (50 – 0.08 μM) in the presence of 10 mM CaCl₂ without preincubation with PADI and quenched at 15 min intervals. Data is normalised to activity of each PADI in the presence of 0.1% DMSO. Data shows mean ± SEM of at least two independent replicates. **C** COLDER assays to determine selectivity of cP4_4 for PADI4 over PADIs 1-3. Apparent IC₅₀ curves were determined at 15-minute intervals with each of PADIs 1-4. Data for the 60-minute time point is shown. Other time points for PADI3 are shown in Figure S11B. COLDER assays were performed at different peptide concentration (50 – 0.08 μM) in the presence of 10 mM CaCl₂ without preincubation with PADI protein. Data is normalised to activity of each PADI in the presence of 0.1% DMSO. Data shows mean ± SEM of at least two independent replicates. **D** COLDER assays to determine K_I and k_{inact} . Apparent IC₅₀ values calculated from COLDER assays displayed in part B with cP4_4 and PADI3 were determined at 15-minute intervals from three independent replicates and the Krippendorff equation was fitted. Data for PADI4 is included from Fig 3C for comparison.⁹

Page 5 lines 116-119. It says there was no warhead hydrolysis or reaction with components of the translation system but it is not explained how this was determined and there is no data to support this claim. It would be good to know if they could confirm lack of reactivity of the electrophile with other proteins.

We have rephrased this section to clarify the evidence we have for the warhead being stable and not interfering with the translation system. See highlights in **green**.

Using this T-stem, similar yields of peptide were achieved as with the control translation suggesting the electrophile-charged tRNA was stable and not interfering with the activity of components in the translation system (Figure S1B). Additionally, no warhead hydrolysis or adduct formation was observed by MALDI-TOF spectroscopy of the translated peptide (Figure S1A).

When the authors introduce the PADI4_3 peptide that was published on BioRxiv by their group, they should cite the sequence and structure shown in figure 2A. I was looking for it for a while before I realized it was not shown until later in figure 2A.

Thank you for bringing this to our attention. We have added the citation of Figure 2A on line 141, indicated in blue highlight, with the first mention of PADI4_3.

The choice of colors for C14 T=0 and T=1h is bad. It is very difficult to see differences in those colors and dotted lines.

The C14 peptide is intended to have the same colours for both timepoints to highlight that it is two different conditions with the same peptide. However, we have changed the line width and point styles to discriminate more clearly between the datasets for 1 hr incubation (T=1) and for no preincubation (T=0), now Figure 4B.

Figure 4: Testing the inhibitory effect of the cP4_4 peptide variants on PADI4.

A The displayed structure of cP4_4 with FAO highlighted in red, and the four modifications made at the FAO position. **B** Inhibition COLDER assays with PADI4 and cP4_4 variant peptides where FAO was substituted with variable groups. COLDER assays were performed at different peptide concentration (50 – 0.003 μM) in the presence of 10 mM CaCl₂. For the reversible warheads, no preincubation between peptide and PADI4 was performed. For Cl4 data is shown

either with (1 hr) or without (0 hr) preincubation of peptide and PADI4. Data is normalised to activity of PADI4 in the presence of 0.1% DMSO. Data shows mean \pm SEM of at least two independent replicates.

Reviewer #2 (Remarks to the Author):

The manuscript titled “Discovering Covalent Cyclic Peptide Inhibitors of Peptidyl Arginine Deiminase 4 (PADI4) Using mRNA Display with a Genetically Encoded Electrophilic Warhead” by the Walport group presents a study on the ribosomal translation of a fluoroamidine warhead that specifically targets cysteine residues. Using FIT system, the authors successfully integrated this warhead into peptide libraries. This approach enabled the de novo discovery of covalent binders and potent inhibitors of PADI4, displaying selective reactivity at the active site cysteine. The findings presented are both novel and significant. The manuscript is well-written, with high-quality characterization and thorough experimentation. I believe that the manuscript is worthy of publication in Communications Chemistry after minor revisions.

Minor concerns:

1. The authors demonstrated the successful ribosomal incorporation of a fluoroamidine warhead in the presence of cysteine, which selectively reacts with the chloroacetyl group without interfering with the warhead. But during the peptide synthesis, the authors initially performed cyclization, and then reacted the cyclized product with ethyl 2-fluoroethanimidate after removing the DDE protecting group to produce macrocyclic peptides containing the fluoroamidine warhead. It would be better to include a more detailed explanation of this process. Is the warhead sensitive to piperidine, or is there another reason?

We agree that it would be helpful to explain our synthetic route and have now extended our discussion of this synthesis in the main text, highlighted in pink.

Initial attempts to selectively deprotect the ornithine side chain using hydrazine to enable warhead addition also resulted in partial removal of the N-terminal Fmoc group. This led to a mixture of peptide products containing both one and two warheads when

ethyl 2-fluoroethanimidate was added. So instead, we devised a strategy where the peptide N-terminus was first deprotected, reacted with N-chloroacetoxysuccinimide, and cyclised with the appropriate cysteine thiol, which was protected with Mmt during the synthesis to allow for selective deprotection. Subsequently, the ornithine side chain was selectively deprotected and reacted with ethyl 2-fluoroethanimidate, before full peptide deprotection, resin cleavage and purification.

2. The authors conducted SPR experiments to assess peptide binding to PADI4. The data analysis used a two-state reaction model, with the reverse rate constant for the second step (k_{-2}) set to zero. This allowed the determination of K_i and k_{inact} values, rather than the classical k_{a1} , k_{d1} , k_{a2} , k_{d2} , and K_D values. A more detailed explanation of this analysis would be beneficial. Additionally, if the SPR results support a covalent binding mode, further discussion on this point may enhance the manuscript.

Thank you for this suggestion. We have now extended our explanation of the SPR experiments in the main text and SI. Changes are highlighted in red.

Main text:

Consistent with the covalent mode of action seen by intact MS and activity assays, during our single cycle kinetics experiments, the baseline of the SPR binding curves did not return to zero, despite elongated dissociation times, indicative of covalent binding. Consequently, a simple 1:1 binding model did not describe the data well. By fitting the data using a two-state reaction model, in which the rate constant for the reverse second step (k_{-2}) was set to zero, the equilibrium constant of the reversible binding step, K_i , could be calculated as the ratio of k_{-1}/k_{+1} and the rate constant of the irreversible chemical step, k_{inact} , as k_{+2} .

SI:

Warhead-containing peptides were analysed using the two-state reaction model, with the rate constant for the reverse second step (k_{-2}) set to zero. The equilibrium constant of the reversible binding step, K_i , was calculated as the ratio of k_{-1}/k_{+1} and the rate constant of the irreversible chemical step, k_{inact} , as k_{+2} . The Arg4, Cit4 and Me4 peptides were fitted using a 1:1 binding model.

The authors might consider including a ChemDraw structure of one or more important macrocyclic peptides containing the warhead to improve readability and comprehension in the main text of the manuscript.

This is an excellent suggestion. We have added in the structure of cP4_4 as part of the new Figure 4, below. We have also displayed the variations we made to this peptide. Changes made are highlighted in the main text in purple.

Figure 4: Testing the inhibitory effect of the cP4_4 peptide variants on PADI4.

A The displayed structure of cP4_4 with FAO highlighted in red, and the four modifications made at the FAO position. **B** Inhibition COLDER assays with PADI4 and cP4_4 variant peptides where FAO was substituted with variable groups. COLDER assays were performed at different peptide concentration (50 – 0.003 μM) in the presence of 10 mM CaCl₂. Data is normalised to activity of PADI4 in the presence of 0.1% DMSO. Data shows mean ± SEM of at least two independent replicates.

4. Did the authors investigate the selectivity of the selected peptides for PADI4 over PADI1, as mentioned in the introduction?

We thank the reviewer for this idea. Please see our response to reviewer 1 as to how we have investigated and shown selectivity. Relevant changes are highlighted in yellow in the manuscript.

All additional renumbering and further changes have been highlighted in grey. We added in Figure 3E which shows what the Krippendorff equation was plotting. Finally, we have also added a link to the repository of sequencing data into the data availability statement.

REVIEWER COMMENTS

Reviewer #1 (Remarks to the Author):

The authors have nicely addressed my concerns, specifically by adding in data for the specificity of lead molecules. The only minor issue is that in the new Fig 3F the symbol key does not match the graph. The colors seem correct but the shapes are not matching with the key.

The symbol key has been updated to match the graphs as below:

Reviewer #2 (Remarks to the Author):

The authors have adequately addressed my concerns, and I recommend publishing it as is.